# Pathogenesis and New Pharmacological Approaches to Noise-Induced Hearing Loss: A Systematic Review

**DOI:** 10.3390/antiox13091105

**Published:** 2024-09-12

**Authors:** Francisco Javier Santaolalla Sanchez, Juan David Gutierrez Posso, Francisco Santaolalla Montoya, Javier Aitor Zabala, Ane Arrizabalaga-Iriondo, Miren Revuelta, Ana Sánchez del Rey

**Affiliations:** 1Otorhinolaryngology Service, Basurto University Hospital, OSI Bilbao-Basurto, BioBizkaia, 48013 Bilbao, Bizkaia, Spain; franciscojavier.santaolallasanchez@osakidetza.eus (F.J.S.S.); gutierrezjuan.orl@gmail.com (J.D.G.P.); francisco.santaolallamontoya@osakidetza.eus (F.S.M.); javieraitor.zabalalopezdematurana@osakidetza.eus (J.A.Z.); 2Otorhinolaryngology Department, Faculty of Medicine, University of the Basque Country, 48940 Leioa, Bizkaia, Spain; ana.sanchezdelrey@ehu.eus; 3Physiology Department, Faculty of Medicine, University of the Basque Country, 48940 Leioa, Bizkaia, Spain; anearriri@gmail.com

**Keywords:** inner ear, noise-induced hearing loss, noise exposure, acoustic trauma, reactive oxygen species, antioxidants, inflammation, anti-apoptotic drugs, anti-inflammatory treatment

## Abstract

Noise-induced hearing loss (NIHL) is responsible for significant adverse effects on cognition, quality of life and work, social relationships, motor skills, and other psychological aspects. The severity of NIHL depends on individual patient characteristics, sound intensity, and mainly the duration of sound exposure. NIHL leads to the production of a reactive oxygen (ROS) inflammatory response and the activation of apoptotic pathways, DNA fragmentation, and cell death. In this situation, antioxidants can interact with free radicals as well as anti-apoptotics or anti-inflammatory substances and stop the reaction before vital molecules are damaged. Therefore, the aim of this study was to analyze the effects of different pharmacological treatments, focusing on exogenous antioxidants, anti-inflammatories, and anti-apoptotics to reduce the cellular damage caused by acoustic trauma in the inner ear. Experimental animal studies using these molecules have shown that they protect hair cells and reduce hearing loss due to acoustic trauma. However, there is a need for more conclusive evidence demonstrating the protective effects of antioxidant/anti-inflammatory or anti-apoptotic drugs’ administration, the timeline in which they exert their pharmacological action, and the dose in which they should be used in order to consider them as therapeutic drugs. Further studies are needed to fully understand the potential of these drugs as they may be a promising option to prevent and treat noise-induced hearing loss.

## 1. Introduction

Noise-induced hearing loss (NIHL) caused by the exposure to high or moderate levels of noise is a common occupational disease [1], with at least 40 million adults in the United States (US) and a prevalence of 13% in the US working population experiencing audiological signs of it in one or both ears [2,3]. NIHL is responsible for significant adverse effects on cognition, quality of life and work, social relationships, motor skills, and other psychological aspects [4,5]. The severity of NIHL depends on individual characteristics, sound intensity, and mainly the duration of sound exposure. Exposure to sounds exceeding 85 decibels (dB), equivalent, for example, to city traffic, for 5 h per week can produce a permanent hearing injury [6]. Moreover, hearing loss may also result from continuous, excessive noise exposure (NE) being the initial mechanism for both acute and chronic NIHL.

Noise is responsible for hair cell (HC) structural damage as well as disrupting connections between inner ear and auditory fibers. Initially, nerve fiber degeneration and hair cell damage occur on both sides of the cochlea, resulting in a 4 kHz notch on audiograms. However, if the damage spreads after NE, it progresses toward the base of the cochlea, leading to medium-to-high-frequency hearing loss [7].

Several theories have been proposed to explain the pathogenesis of NIHL, but the evidence demonstrates that the main mechanisms underlying NIHL result in cell inflammation, increased oxidative stress due to reduced cochlear blood flow, reactive oxygen species (ROS) production, and elevated intracellular calcium levels [1,8,9]. The impaired signaling cascade provokes alterations in the mitochondrial function and triggers the activation of apoptotic pathways, DNA fragmentation, and cell death [10,11,12,13,14].

### 1.1. Effect of ROS

Sensory hair cells (HCs) are especially sensitive to high levels of oxidative stress owing to the high metabolic energy required for the sound transduction mechanism [15,16]. Since HCs do not regenerate naturally, this can lead to irreversible NIHL. Exposure to extremely loud noise can result in an excess of free radicals within cells, including ROS, reactive nitrogen species (RNS), and lipid peroxides [1,8,17,18]. At this level, a proper antioxidant system, including enzymes like glutathione (GSH), GSH peroxidase, or superoxide dismutase, are essential for maintaining a balance in cell redox revels [12]. Without these defenses, ROS accumulation can produce cell inflammation, cell damage, or even the cell death of HCs and auditory neurons.

### 1.2. Effects of Mitochondrial Dysfunction and Elevated Intracellular Calcium

Mitochondria are involved in metabolic damages caused by acoustic trauma. The dysregulation of intracellular Ca^2+^ levels is responsible of impaired mitochondrial function. The mechanism underlying elevated intracellular Ca^2+^ levels is due to an overexpression of a mitochondrial calcium uniporter (MCU) in HCs after intense NE. This MCU is the major calcium channel for Ca^2+^ influx from the cytosol into the mitochondria, so acoustic trauma elevates intracellular Ca^2+^ concentrations and consequently stimulates mitochondrial chain reactivity and ROS production [19,20].

### 1.3. Effect of Apoptotic Cascade Activation

Apoptosis is associated with DNA nuclear fragmentation and cell death. The apoptotic cascade is controlled by several genes and proteins, including caspases. In HCs, caspases 3, 8, and 9 are activated after traumatic sound exposure, provoking physical HC damage. The activation of these caspases leads to mitochondrial membrane potential alteration, the formation of apoptosome, cytochrome c release, and membrane lipid peroxidation [21]. The precise molecular mechanism leading to HC death is unclear, however several authors highlight the roles of protein kinases, like stress-activated protein kinase (SAPK), c-jun N-terminal kinase (JNK), and mitogen activated protein kinase (MAPK), in the activation of the apoptotic cascade [22].

### 1.4. Antioxidant Treatment of the Disease

Although there is not much evidence of the effects of antioxidants to prevent or minimize the progression of NIHL in humans, some authors suggest that exogenous antioxidant treatment can be a therapeutic approach to reduce the progression of hearing loss [1,23]. Antioxidant therapies have been postulated as a strategy to reduce age-related hearing loss in several animal models [24]. Other authors confirmed that the combination of several antioxidants could potentiate their protective role against NIHL [25]. Nevertheless, there have been limited studies that examine the specific mechanisms of antioxidants and their potential therapeutic benefits for improving, delaying, or preventing NIHL. As a result, this study seeks to conduct a comprehensive review of the antioxidants and other pharmacological therapies that have been studied, and the ways in which they may help in treating NIHL.

### 1.5. Other Pharmacological Treatments for the Disease: Anti-Inflammatory and Anti-Apoptotic Treatments

As previously mentioned, inflammation and apoptosis also play an important role in NIHL. Exposure to noise activates nuclear transcription factor-Kappa β (NF-Kβ) in the cochlea, leading to an increase in pro-inflammatory cytokines, caspases, and pro-apoptotic molecules that promote hearing loss [26]. Several anti-apoptotic and anti-inflammatory molecules, such as glucocorticoid, have been proposed as potential therapeutic targets to prevent NIHL. Among these anti-inflammatory products, several natural substances have been studied in order to reduce HC degeneration. Consequently, this review aims to thoroughly investigate these new and potential treatments that reduce NIHL.

Logically, noise prevention and ear hygiene are always effective and efficient strategies to deal with NIHL, especially in people professionally exposed to high levels of noise. However, in many cases, such as armed forces personnel or people exposed to explosives or firearms, this prevention is impossible to achieve effectively. This is why the study of the pharmacological therapies for the treatment of NIHL is of great interest.

## 2. Materials and Methods

To write this systematic review, the guidelines of Preferred Reporting Items for Systematic Reviews and Meta-Analyses (PRISMA) were used [27]. An updated search of the main databases available was conducted in order to achieve the best evidence-based knowledge.

### 2.1. Search Strategy

a.For antioxidants:

PubMed/Medline (Mesh): (“Hearing Loss, Noise-Induced” [Mesh]) AND “Antioxidants” [Mesh].

Cochrane Library and Embase: ((antioxidant*) OR (“dietary supplement*”) OR (N-Acetylcysteine) OR (NAC)) AND ((“Hearing Disorders”) OR (“Acoustic Trauma”) OR (“noise induced hearing loss”) OR (NIHL) (noise exposure).

Science Direct and Google Scholar: “noised induced hearing loss” AND “antioxidants”.

b.For other pharmacological treatments:

PubMed/Medline (Mesh): (“Hearing Loss, Noise-Induced” [Mesh]) AND “Drug Therapy” [Mesh].

Cochrane library, Science Direct, and Web Of Science: “Noised induced hearing loss AND pharmacological treatments".

### 2.2. Eligibility Criteria

Inclusion criteria:(1)Studies related to antioxidants, anti-inflammatory, and/or anti-apoptotic drugs.(2)Studies performed on adult individuals (older than 18 years old).(3)Studies on experimental animals.(4)Studies published before January 2024.

Exclusion criteria:(1)In vitro studies.(2)Studies that were not performed with antioxidants, anti-inflammatory, and/or apoptotic drugs.(3)Studies on children and adolescents.(4)Studies on patients with previous otologic pathologies.(5)Studies that were not written in English.(6)Inaccessible articles.

After the initial search of the databases and removal of duplicated articles, the screening of titles and abstracts was conducted by two independent researchers (FJ.S. and A.S.) to exclude irrelevant articles as well as those that did not meet the inclusion criteria. The full texts of the remaining related articles were then carefully evaluated by three researchers (FL.S., F.S., and A.S.) to select appropriate articles based on the methodology and results. Any inconsistency between the researchers was resolved by consulting with three other researchers (JD.GP, A.Z and MR).

Two independent researchers (FJ.S. and JD.GP.) extracted the following information from the selected studies: author’s name, study location, study design, study population, mean age, sex, sample size, type, dose, duration of intervention (clinical trials), control group, serum vitamins/antioxidants (cross-sectional and cohort studies), and outcomes.

### 2.3. Search Strategy of the Applicability of the Pharmacological Therapies in Human Beings

Once the studies included in this systematic review were examined and analyzed, a third search was performed with the objective of elucidating the degree of applicability of the pharmacological therapies included in this review for human beings.

This additional search consisted of analyzing whether the pharmacological therapies used in the 41 included studies had any pharmacological indications approved by the Food and Drug Administration (FDA) and the European Medicines Agency (EMA) for human use. In addition, a search in PubMed was added to the pharmacological therapies that received any approval by the FDA-EMA under the following terms “[(pharmacological therapy) AND (Noise Induced Hearing Loss)]” with the aim of examining the existence of RCTs and/or cohort studies conducted on human beings.

## 3. Results

### 3.1. Study Selection

As previously mentioned, we stratified the search in two levels: the first one was specifically directed to find the antioxidants described for the treatment of NIHL, and the second one had the objective of finding the rest of the existing pharmacological therapies (anti-inflammatory and anti-apoptotic drugs). We decided to tackle the search via this strategy because, in the preliminary searches, we realized that the pharmacological treatments described in the literature for NIHL could be subdivided into three main categories:(1)Antioxidants.(2)Anti-inflammatory drugs (glucocorticosteroids and non-corticosteroids).(3)Anti-apoptotic drugs.

Using double-search strategies, the initial database search provided 7963 articles. First, (n = 4704) articles were removed due to duplications and/or the matched articles were not of interest in the following revision. In the second phase, the titles and abstracts of the remaining articles (n = 3259) were checked. (n = 3056) articles were excluded by means or inclusive or exclusive criteria. Finally, in the third phase, there were (n = 203) articles left. Among these 203 articles, (n = 162) articles were excluded because they were considered outdated and/or there were many more recent students concerning the antioxidant analyzed. Therefore, we did not include articles published before 2006 in order to keep this systematic revision up to date. In the end, (n = 41) articles were included in the systematic revision. Flowcharts of the selected studies and the progression for this selection are shown in Figure 1 and Figure 2.

### 3.2. Study and Participant Characteristics

Among these studies, five were clinical trials [28,29,30,31,32], 35 were prospective observational studies, and 1 a retrospective cohort. Four studies were carried out on humans: one on personnel in Swedish armed forces [33], two on South Korean armed forces [34], another on Iranian textile workers [29], and the last one on people exposed to gunshot fire [35]. The characteristics of the included studies are presented in Table 1, Table 2, Table 3 and Table 4.

Taking all this into consideration, the otoprotective therapies with more op-to-date evidence, to our knowledge, are presented in the tables below.

### 3.3. Antioxidants

Antioxidants have been suggested to be crucial for the protection of the inner ear by decreasing the production of ROS/RNS and promoting cochlear vasodilation. The most outstanding findings of this systematic review concerning antioxidants are shown in Table 1.

The main results shown in Table 1 are presented below.

**Table 1 antioxidants-13-01105-t001:** Characteristics of the antioxidants that were included in the systematic review and their main outcomes.

References	Design	Population	Age	Gender	Treatment	Noise Exposure	Type, Dose, and Duration of Intervention	Parameters Analyzed
1. Campbell (2023). Springfield (US) [36]	Prospective cohort	Chinchilla lanigera	3–5 years	male	D-methionine	a. Steady-state noise (105 dB SPL at 4 kHz for 6 h) b. Impulse noise (155 dB 150 times × 75″)	D-met dose levels 0, 50, 100, or 200 mg/kg/dose	a. ABR (t = 0); t (+24 h). b. Serum (GPx, Gr, SOD, and CAT) c. Cochlear enzymes (GSH/GSSG)
2. Campbell (2021). Springfield [37]							D-met 200 mg/kg (5 ip injections c/12 h for 48 h)	a. ABR (t = 0); t (+21 d). b. Serum (GPx, Gr, SOD, and CAT) c. Cochlear enzymes (GSH/GSSG)
3. Campbell (2022). Springfield [38]							D-met dose levels 0, 50, 100, or 200 mg/kg/dose	a. ABR (t = 0); t (+21 d). b. Serum (GPx, Gr, SOD and CAT) c. Cochlear enzymes (GSH/GSSG)
4. Rosenhall (2019). Gothenburg (Sweden) [33]	Retrospective cohort	Personnel in Swedish armed forces	22.9 years	both	NAC	a. Small-caliber weapons (77%) b. Large-caliber weapons (7%)	NAC 400 mg	a. PPTA 1 (pre-exposure) b. PPTA 2 (4–72 h post-exposure) c. 9 months post-exposure
5. Ada (2017). Kirklareli (Turkey) [39]	Prospective cohort	Wistar albino	-	male		1–12 kHz band white noise; 110 dB for 6 h	NAC 100 mg/kg/day by gavage × 7 d	a. Light microscopic b. Scanning electron microscopy
6. Choi (2014). Oklahoma City (US) [30]	RCT	Chinchilla lanigera	3–5 years	female	NAC + 4-OHPBN oral	4 kHz band noise; 105 dB for 6 h	NAC + 4-OHPBN oral 4 h post-noise c/12 h × 2 d	a. ABR (pre-exposure; immediately after exposure; t = +21 d) b. DPOAE c. PPTA d. Scanning electron microscopy
7. Doosti (2014). Tehran (Iran) [29]	RCT	Textile workers	39 years	male	NAC vs. ginseng	continuous noise > 85 dB for 8 h a day	1. NAC 1200 mg/day 2. Ginseng 200 mg/day	a. PPTA: pre-exposure and +15 d
8. Lu (2014). Oklahoma (US) [28]	RCT	Long-Evans and Sprague Dawley	-	male	NAC/HPN-07	OBN 10–20 kHz (115 dB) 1 h	Control: 5 mL/kg saline Case: NAC/HPN-07 5 mL/kg (60 mg/mL) intraperitoneally. Administration: +1 h; +2 d (×2)	1. Plasma levels of total free cysteine and HPN-07 2. ABR; TS; DPOAE; LS a. 3–5 d before treatment b. 8 h, 24 h, 7 d, and 21 d after treatment
9. Kilic (2022). Erzurum (Turkey) [31]	RCT	Sprague Dawley	-	female	Berberine	4 kHz white noise 110 dB for 12 h	100 mg/kg berberine single dose daily × 5 d (intragastric lavage)	a. DPOAE (before treatment; t = 6 d) b. Histopathological and immunohistochemical
10. Zhao (2021). Xuzhou (China) [40]	RCT	Guinea pigs	10–12 weeks	male		6.3–20 kHz white noise 120 dB for 4 h (2 h noise, 8 h rest, and 2 h noise)	5 mg/mL nanoparticle left in the RWM	a. ROS-scavenging ability (SOD and MDA) b. Inflammatory cytokines in cochlea (TNF-α, IL-6, and IL-1) c. Morpho-functional study (immunofluorescence t = +12 d and SEM) d. ABR 4, 8, 16, and 24 kHz (t = −2 d, 0 d, 2 d, 4 d, 6 d, 8 d, 10 d, 12 d, and 14 d) e. Toxicity: H&E staining t = 14 d (heart, liver, spleen, and kidney)
11. Xiong (2017). Guangzhou (China) [41]	Prospective cohort	C57 BL/6 mice	2 months	female	Resveratrol	10 kHz, 120 dB for 1 h.	Resveratrol 4 g/kg for 2 months (300 mg/kg per day)	a. ABR (before treatment, end of treatment, immediately after noise exposure, and + 15 d)
12. Seidman (2014). Michigan (US) [32]	RCT	Fischer 344	2–3 months	male		Broadband noise, 105 dB for 24 h	4 mg/kg resveratrol daily for 3 d by gavage	a. Cox-2 (Western blot) b. ROS in lymphocytes (fluorescence flow cytometry histogram)
13. Chen (2020). Buffalo (US) [42]	Prospective cohort	Sprague Dawley	10 months	-	HK-2	95 dB sound pressure level (SPL) for 8 h/d for 21 days	0.2 wt. % HK-2 corresponding to an oral dose of 125 mg/kg/day HK-2	Compound action potential (CAP). The CAP, which reflects the gross neural output of the cochlea, was recorded for all animals approximately 2 months post-exposure
14. Fetoni (2018). Rome (Italy) [43]	Prospective cohort	Wistar	adult	male	Rosmarinic acid	60 min to a 120 dB SPL	20 μL RA solution injected into the tympanic bulla 1 h before noise exposure RA solution (10 mg/kg) injected 1 h before noise exposure; once daily for the following 3 days	ABR assessed bilaterally before noise exposure to ensure normal hearing and reassessed at 1, 3, 7, 15, and 30 days after noise exposure
15. Richter (2018). Chicago (US) [44]	Prospective cohort	Guinea pigs	no information	-	Fluvastatin	4–8 kHz, 120 dB re 20 μPa, for 4 h	Fluvastatin was implanted in the guinea pig’s left cochlea The pump delivered 50 μM	Frequency measured (2, 4, 8, 16, and 32 kHz)
16. Zhang (2022). Beijing (China) [45]	Prospective cohort	Wistar	2 months	male	SOD ZIF-8	120–125 dB for 12 h/day for 3 days	10 μL of SOD ZIF-8; 2 mg/mL suspension	ABR in rats before drug administration, and at 1, 3, 7, 14, and 28 days after noise exposure
16. Lee (2013). Oklahoma (US) [46]	Prospective cohort	CBA mice	4 weeks	-	Ginkgo biloba	BWN (110 dB SPL) for 1 h	Gingko biloba administered once a day for 7 days before noise exposure; 3 mg/kg powder	ABR and DPOAE
17. Fetoni (2012). Rome (Italy) [47]	Prospective cohort	Wistar	2 months	-	Q10	100 dB (SPL) × 10 consecutive days × 60 days	Group II: Q-ter (100 mg/kg) Group III: 20 µL Q-ter Group IV: 40 µL Q-ter	ABR at low (6, 12 kHz), mid (16, 20 kHz), and high (24, 32 kHz) frequencies ABR before noise exposure; 1, 3, 7, and 21 days after noise exposure
18. Sergi (2006). Rome (Italy) [48]	Prospective cohort	Hartley albino guinea pigs	3 months	-	IDB	Pure tone of 6 kHz for 40 min to a 120 dB SPL	Intraperitoneally: 5 mg/kg 1 h before noise exposure and once daily for the following 3 days	Electrophysiological tests: before, and +1 d, +7 d, and +21 d after noise exposure
19. Fetoni (2008). Rome (Italy) [49]	Prospective cohort		3 months	-		40 min to a 6 kHz, 120 dB SPL	IDB (intraperitoneally: 5 mg/kg) Vitamin E (intramuscularly: 1360 IU/g) Dissolved in corn oil (10 mg/mL at 100 or 50 mg/kg body weight)	ABR measured before noise exposure, and 1 h, 3 days, 7 days, and 21 days after noise exposure

Abbreviations: RCT (randomized controlled trial); PTS (permanent threshold sift); TTS (temporal threshold sift); DPOAE (distortion product otoacoustic emission); ABR (auditory brainstem response); NAC (N-Acetylcysteine); Hz (Hertz); kHz (kilohertz); dB (decibels); PTTA (pure-tone threshold audiometry); RWM (round window membrane); HK-2 (1-(5-hydroxypyrimidin-2-yl) pyrrolidine-2,5-dione); IDB (idebenone); GPx (glutathione peroxidase).

#### 3.3.1. D-Methionine (D-Met)

Campbell et al. (2021) studied the relationship between D-methionine and NIH [37]. Between 2021 and 2023, this group published three articles that analyzed the protective effects of this antioxidant. They studied a prospective cohort of male chinchilla lanigera that were exposed either to steady-state noise (105 dB SPL at 4 kHz for 6 h) or to impulse noise (155 dB 150 times × 75″) under different conditions prior to or after D-met administration.

In the first article [37], they concluded that, in contrast to previous evidence, D-met rescue did not reduce permanent or temporary threshold shifts. However, they demonstrated that D-met rescue altered selective serum and cochlear oxidative state changes 24 h post-noise related to the control group.

In the second study conducted by this group [38], they analyzed the otoprotective effect of the preloaded administration of D-met. They determined that the most effective time window for preloading D-met was from 3.5 to 2.5 days before exposure to continuous noise, and from 3 to 2 days before exposure to impulse noise. The study also showed that, 21 days after exposure to sudden noise, D-met increased the levels of serum GR at two specific preloading time points, while also increasing serum SOD and GPx levels for continuous NE. However, GSSG levels and the GSH/GSSG ratio were not affected by D-met under any circumstances.

In their third study [36], the researchers found that D-met significantly reduced hearing loss caused by impulse noise exposure (at doses of 100 mg and 200 mg) and continuous noise exposure (across all doses, time frames, and frequencies tested). They also observed an increase in serum SOD levels (at doses of 100 mg and 200 mg for 24 h following exposure) and GPx levels (at a dose of 50 mg/kg for 24 h following exposure) 21 days after NE. These results led the researchers to conclude that the wide range of effective doses and timing of D-met administration make it a promising antioxidant for hearing protection.

#### 3.3.2. N-Acetyl-L-Cysteine (NAC)

Lu et al. [28] conducted a randomized controlled trial (RCT) in order to analyze the otoprotective effects of NAC/HPN-07, a combined treatment of an antioxidant (NAC) and the free radical reagent disodium 2,4-disulfophenyl-N-tert-butylnitrone (HPN-07). The ABR of the treatment groups showed that NAC/HPN-07 substantially reduced auditory thresholds at all test frequencies (2–16 kHz), beginning 24 h after NE and continuing for up to 21 days.

Similarly, Doosti et al. [29], in their study conducted on Iranian textile workers, demonstrated that there was a noise-induced threshold shift reduction in both ears for the NAC and also for Ginseng groups at 4, 6, and 16 kHz (*p* < 0.001). Nevertheless, they observed that the otoprotective effects were more prominent in the NAC group than in the ginseng one.

Rosenhall et al. [33] in (2019) conducted an observational study on the personnel of the Swedish armed forces, in which they analyzed the otoprotective effect of NAC at 400 mg when administered after weapon fire. They showed that the occurrence of hearing thresholds ≥ 25 dB HL and the development of threshold shifts ≥ 10 dB were less frequent in the NAC group compared to the non-NAC group post-NE.

In 2014, Choi et al. [30] analyzed the effect of NAC + nitrogen-based free radical (4-OHPBN) administered orally to female chinchilla lanigera. They concluded that NAC + 4-OHPBN significantly reduced the DPOAE threshold shift and percentage of missing outer hair cells (OHCs) in a dose-dependent manner. In a study conducted by Ada and colleagues in 2017 [39], they observed that the use of NAC in the acoustic trauma (AT) group led to a decrease in damage with preserved cochlear structures if observed through light microscopy.

#### 3.3.3. Berberine

Zhao et al. [40] (2021) conducted a study in which they analyzed the otoprotective effect of berberine in guinea pigs exposed to 6.3–20 kHz white noise at 120 dB for 4 h. In this article, the authors compared the distribution of berberine within the OHCs depending on the nanoparticles used to distribute it. In the analysis by groups, they demonstrated that PL-PPS/BBR, a Prestin-targeting peptide 2 (PrTP2)-modified nanoparticle, successfully reduced ROS and inflammation, preventing further OHC damage after NIHL.

On the other hand, Kilic et al. [31] conducted an RCT in 2022, in which they analyzed DPOAEs and performed histopathological and immunohistochemical analyses of Sprague Dawley rats after AT exposure. They observed that DPOAEs showed a significant decrease at higher frequencies in the trauma group compared to the other groups.

#### 3.3.4. Resveratrol

Seidman et al. [32] analyzed the antioxidant effect of resveratrol on 344 Fischer male rats exposed to AT. In their RCT, they compared ROS production and the scavenging effect of resveratrol administered after AT exposure. They observed an upregulation of COX-2 protein expression after AT exposure, but this upregulation was diminished after resveratrol administration in blood samples from the AT group compared to the control ones.

Similarly, Xiong et al. [41] demonstrated in 2017 that a long-term diet supplemented with high-dose resveratrol to C57BL/6 mice resulted in increased cochlear Sirtuin 1 (SIRT1) activity and reduced HC loss after NE compared to the control group, concluding that resveratrol feeding diminishes ROS production in the cochlea after NE.

#### 3.3.5. 1-(5-Hydroxypyrimidin-2-yl) Pyrrolidine-2,5-dione (HK-2)

Chen et al. [42] administered different doses of an HK-2 antioxidant after NE to Sprague Dawley rats. Higher doses (40 mg/kg/d) of HK-2 provided greater protection than lower ones (16 mg/kg/d) compared to the control group. Moreover, they confirmed that HK-2 administration after NE reduces OHC lesions.

#### 3.3.6. Rosmarinic Acid (RA)

Fetoni et al. [50] used 226 adult Wistar rats assigned to six experimental groups in an RCT. They injected the RA solution (10 mg/kg) 1 h before NE once a day for the next three days. They compared the results to a control group plus RA, total exposure for 4 days without noise, and to a control group with saline administration. Auditory function and histological assessments suggest that RA attenuates a threshold shift as well as OHC and IHC losses after NE. At the immunohistochemical level, RA reduced superoxide production and lipid peroxidation. It induced the activation and translocation of Nrf2 in the spiral ganglion neuron nucleus. The enhancement of Nrf2 expression after RA administration also appeared to induce the upregulation of HO-1 protein expression.

The same author in another study [43] administered intrathoracic RA to 58 adult Wistar rats proving its otoprotective effect, with a threshold shift of approximately 25–30 dB in the first post-treatment day. After NE, morphological damage mainly involved the outer row of OHCs, and only a few IHCs were lost. 

#### 3.3.7. Statins

Richter et al. [44] demonstrated surprisingly that the administration of Fluvastatin via an osmotic pump into one ear protects the contralateral ear after noise exposure with HC protection. However, this protection was not consistent in all the studied animals, so further studies must be performed in order to determine this mechanism. These studies were corroborated with others, such as the ones performed by Park et al. [51] that reported a 20 dB ABR threshold shift decrease in animals treated with pravastatin compared with non-treated animals after NE. This was confirmed with HC protection, suggesting that statins may induce protection after NE. 

#### 3.3.8. SOD ZIF-8

Zhang et al. [45] used superoxide dismutase SOD treatment, included in a metal organic framework (ZIF-8), as treatment for NIHL. For this purpose, they used Wistar rats exposed to broadband white noise in the range of 120–125 dB for 3 consecutive days for 12 h/day. They observed that SOD-ZIF-8-treated animals had improved hearing at mid–high frequencies in particular, and that they had higher hair cell survival in the middle and basal turns.

#### 3.3.9. Ginkgo Biloba

Lee et al. [46] performed an experiment using 4-week-old C57BL/6 mice that received the administration of Ginkgo biloba 7 days after NE. Ginkgo biloba has a flavonoid that improves blood flow and oxygen supply by reducing oxidative stress. In this study, they observed that there was a hearing level threshold increase after Ginkgo biloba administration in the NE group, with better DPOAE results. 

#### 3.3.10. Coenzyme Q10

Fetoni et al. [47] used adult Wistar rats that were administered a Q-ter regimen using different administration methods and doses. They performed acoustic trauma using a waveform generator and produced a continuous pure tone at 10 kHz. They observed that animals with AT and Q-ter administration had less HC loss and less lesions in the same region.

#### 3.3.11. Idebenone

Sergi et al. [48] used adult Hartley albino guinea pigs that were administered idebenone intraperitoneally, an NADPH cofactor that reduces oxidative stress. First, they produced AT with a waveform generator and created a continuous pure tone at 6 kHz for 40 min and 120 dB sound pressure. After this, they counted in the basal, second, third, and apical turns of the cochlea the number of OHCs and IHCs. Idebenone-treated group showed better preservation of these HCs from acoustic trauma with less OH loss. 

Another study performed by Fetoni et al. [4] compared the administration of Idabenone combined with vitamin E in albino guinea pigs after sound exposure, and corroborated the previous results with better hair cell preservation. 

### 3.4. Anti-Inflammatory Drugs

#### 3.4.1. Glucocorticosteroids 

The success of glucocorticoid (GC) therapy for NIHL consists of the regulation of cochlear homeostasis, in which glucocorticoid receptors (GRs) play a fundamental role. In the genomic pathway, GCs bind to GRs in the cytoplasm and induce a conformational change that results in the binding of specific DNA sequences to target genes in the nucleus. This will activate or inhibit the transcription of target genes involved in inflammatory, oxidative, and apoptotic processes [52]. GCs are supposed to stop this chain of events, maintaining the functional activity and histological structure of the cochlear organ [53]. The main results obtained in the review are presented in Table 2, and the most representative results are explained as below.
antioxidants-13-01105-t002_Table 2Table 2Characteristics of anti-inflammatory corticosteroids that were included in the systematic review and their main outcomes.ReferencesDesignPopulationAgeGenderTreatmentNoise ExposureType, Dose, and Duration of InterventionParameters AnalyzedMain Outcomes1. Panevin (2018). St. Petersburg (Russia) [53]Prospective cohortWistar-MaleHydrocortisone acetate suspension5 kHz (110–112 dB) for 2 hIV 0.05% hydrocortisone acetate suspension (5 mg/kg) + 0.01% lidocaine hydrochloride solution (1 mg/kg)DPOAE: −1 d and +1 h; +24 h +7 dThere was a partial recovery of the DPOAE amplitude at 4 kHz, 1 h after the injection of hydrocortisone. The same status was recorded only by day 7 in response to the hydrocortisone solution. The functional activity of the acoustic receptor recovered completely in response to the hydrocortisone suspension during this period for all checkpoints of OAE. The detected differences could be attributed to the presence of the dispersed drug (povidone particles) that contained the hydrocortisone suspension2. Muller (2016). Heidelberg (Germany) [54]Prospective cohortGuinea pig--Prednisona (PD) or methylprednisolone (MTP)Impulse noise bursts (500 ms) at (0.25–4 kHz; 140–144 dB SPL); 15, 30, 45, 60, and 120 bursts were carried outOsmotic pump implanted in the RWM delivering glucocorticoids (0.5 μL/h) for 2 weeks: PD (25 mg/mL) or MTP (12.5 mg/mL)Hearing threshold measured using the compound action potential (CAP): 0 d and +14 d At 15–30 impulse noise bursts, there was great variability in the threshold and partial loss of OHCs At 60–120 impulse noise bursts, the loss of threshold and hair cells was complete. With 45 impulse noise bursts, a loss of 75% of hair cells was observed. A reduction in hearing loss was observed after 2 weeks of high exposure to PD at 25 mg/mL without a significant reduction in the percentage of damaged OHCs. With the use of MTP at 12.5 mg/mL, a reduction in hearing loss and a decrease in the loss of OHCs were observed at 45 impulse noise bursts 3. Chang (2017). Yangju (South Korea) [35]Prospective cohortHumans21–22 yearsMalePD and Dexamethasone (DEX)Gunshot noise (149 dB) measured at 1 m; 20 shots per session, 3–5 times sequentially Oral PD (60 mg) daily for 10 d with tapering for 4 d + oral Ginkgo biloba (40 mg) twice a dayITSI DEX (5 mg/mL) in 4 applications every 2 daysPure tone air conduction threshold audiometry: 0 d, +1 monthGroup 2 (ITSI group) showed a significant improvement compared to group 1 (PD group) at all frequencies 4. Choi (2019). Ansan-si (South Korea) [34] Retrospective cohort20–21 yearsPDGunshot noise exposureOral Prednisone (60 mg)Pure tone thresholdsThe post-treatment hearing threshold of group 1 showed a significant improvement (*p* < 0.05) while group 2 did not. The regression coefficient for the association between the initial hearing level and the hearing gain was 0.45 5. Heinrich U-R (2016). Mainz (Germany) [55] Prospective cohortGuinea pig2 weeksDEX90 dB noise exposureIT DEX (0.1 mL) 4 h before NE ABRs: 0 d, +2 hThere was a significant reduction (*p* = 0.0013) in the mean hearing thresholds of the DEX group (23 dB) compared to the NE (31.99) and saline (28.5) groups at the end of the treatment. The increase in staining intensity in GR expression was only significant for the DEX group (*p* = 0.0344). Strong associations were detected when comparing the GR expression in the fibrocytes of the limbus with the spiral ganglion cells, the interdental cells, or the nerve fibers (*p* < 0.001)6. Harrop-Jones (2016). California (US) [56]6–8 weeksDEX in poloxamer vehicle (OTO-104)Narrowband noise (4–8 kHz at 105–110 dB) for 2 hIT bilateral OTO-104 prior to NEABRa. Doses of 2% and 6% OTO-104 exhibited otoprotection against NIHL b. But only the intratympanic injection of 6% OTO-104 showed significant otoprotection after NIHL7. Gumrukcu (2017). Istanbul (Turkey) [57]WistarAdultsFemaleDEX110 dB for 25’IT DEX 1 mL (4 mg/mL) on days 0, 2, and 4 DPOAE: 0 d, +7 d, and +10 dDEX treatment group showed a significant difference in DPOAE measurements on days 7 and 10 at all frequencies compared to the saline group8. Zhu C (2018). Viena (Austria) [58] Guinea pigAdultsBothDEX and triamcinolone acetonide (TAAC) in thermoreversible poloxamer 407 (POX-407) hydrogels120 dB SPL for 3 h Hydrogels applied post-NE into the RWM.ABR: −1 week and +1 d, +2 d, +7 d, +14 d, +21 d, and +28 da. DEX (6%) group showed significantly higher threshold shifts than the TAAC (6%) group on day 14 and exhibited a significantly smaller hearing threshold shift at 16 kHz compared to the control hydrogel on day 7 b. SGC showed the highest density on the second turn in the DEX (6%) group, but no statistically significant difference was detected9. Shih (2018). Taipei (Taiwan) [59] AdultsMaleDEX +USMB118 dB SPL for 5 h IT DEX/MB US or DEX/MB without US. US irradiation: 3 w/cm^2^ for 3 consecutive 1’ course duty cycle 50%. In the MB group, they used 2 preparations: one called SonoVue made of phospholipids, and the other with albumin.ABR: 0 d and +1 d, +7 d, +14 d, +28 da. DEX concentration was significantly higher in the US MB SonoVue groups with frequencies of 0.5 MHz, 3 MHz, and 5 mhZ than in the RWS group. Albumin MBs also significantly increased the perilymphatic DEX levels in the USM group compared to the RWS b. DEX delivery was significantly higher for a US frequency of 1 MHz than for 0.5 MHz US group showed significant differences on days 7, 14, and 28 than the control group 10. Park (2020). Seul (South Korea) [60]Sprague Dawley--DEX in different vehicles: saline, hyaluronic acid (HA), and methoxy polyethylene glycol-b-polycaprolactone block copolymer (MP)120 dB SPL for 3 h IT DEX (10 mg) + MP; 3 h NE a. ABR: 0 d, +1 h NE, +3 h after vehicle/drug injection, and +4 d, +8 d, +30 d, and +45 d after IT b. Endoscopic examination TM c. Microcomputed tomography: +2 h, +4 h, +8 h, +30 h, and +45 h after ITAll 4 groups improved the hearing threshold after treatment, but there was a difference in degree. The HA + D group was significantly better than the MP + D group (*p* = 0.043). The difference between the HA + D group and the saline + D group was not statistically significant (*p* = 0.083). The duration of vehicle/drug in the bulla was significantly longer for the MP + D group than the saline + D and HA + D groups (*p* = 0.038) 11 Jeong (2021). Seul (South Korea) [61]7–8 weeksMaleDEX and sodium caprate (SC)115 dB SPL for 5’ IT DEX sodium phosphate (5 mg/mL) + Sodium caprate (1.94 mg/mL) ABRa. Perilymphatic DEX concentration was significantly elevated in the DEX + SC group compared to the DEX-alone group: at 30’ post-NE (*p* = 0.0255) and at 90’ (*p* = 0.0206) b. SC co-treatment accelerates the recovery of NIHL c. SC did not show any signs of loss or death of IHCs and OHCs 14 days after treatment

##### Hydrocortisone

Panevin and Zhuravskii [53] studied the effect of intravenous hydrocortisone (in solution and in suspension with povidone) on Wistar rats exposed to AT. The results showed that hydrocortisone in suspension had a greater otoprotective effect 1 day after the acoustic stimulation with respect to the solution. That could be explained by the size of the transport particle (povidone), in the range of 100–500 nm, which has a direct relationship called ototropism, given that it has been observed that the blood/cochlear labyrinth barrier is permeable to substances of sizes between 160 and 280 nm.

##### Prednisone (PD)

In 2016, Muller et al. [54] studied the use of Prednisone (PD) and methylprednisolone (MTP) administered directly into the cochlea through the RWM in guinea pigs. In their study, they demonstrated a reduction in hearing loss after 2 weeks of treatment with PD at 25 mg/mL. However, even if they observed an improvement in the percentage of damaged OHCs, this was not significant. They also determined that MTP at 12.5 mg/mL caused a reduction in hearing loss and a decrease in the loss of OHCs.

On the other hand, Chang et al. [35] studied in a human cohort the therapeutic use of systemic PD (60 mg per day for 10 days) and the simultaneous combination with IT Dexamethasone (DEX) in 4 doses of 5 mg/mL every 2 days. This study resulted in a significant improvement in audiometric results after the acoustic trauma produced by military firearms training at a maximum noise level.

In the same way, in 2019, Choi et al. [34] conducted a study carried out on military personnel, in which they demonstrated that systemically administered PD had a positive effect on hearing recovery after a traumatic blast. They observed that the administration of a high dose of PD (60 mg per day) for 10 days and a reduced dose for the remaining 4 days were optimal to obtain a greater recovery result for acute acoustic trauma.

##### Dexamethasone (DEX)

Heinrich et al. [55] demonstrated that the IT administration of DEX 14 h before NE increases GR expression in fibroblasts, spiral ganglion cells, and interdental cells, thus stabilizing the function of the inner ear.

The action time of a drug administered into the middle ear is short because its contact with the RWM is limited due to its drainage through the Eustachian tube. In order to solve this problem, Harrop-Jones et al. studied the combined administration of DEX with a linked poloxamer hydrogel: OTO-104. In their study, they demonstrated an otoprotective effect at concentrations of 2% and 6% in an NIHL guinea pig model. They also observed that the permanent threshold shift (PTS) and temporary threshold shift (TTS) were improved after OTO-104 application [56].

In 2018, Zhu et al. [58] also observed accelerated hearing recovery in a guinea pig model after the use of a Poloxamer hydrogel with DEX (6%). On the other hand, they studied that triamcinolone acetonide (TAAC) at 6% and 30% with DEX provided no otoprotection.

Another vehicle for GC administration into the inner ear is air-core microbubbles (MBs) wrapped in a layer of lipids, proteins, or polymers using sonophoresis with ultrasound (US). In the study carried out by Shih et al. [59], they demonstrated that MB with lipid or protein preparation (albumin) presented a higher level of perilymphatic concentration of DEX in the US group compared to the IT application of DEX alone on days 7, 14, and 28 post-NE. In addition, they observed a preservation of OHCs in the basal and middle cochlear turns.

Hyaluronic acid (HA) and the methoxy block copolymer polyethylene glycol-b-polycaprolactone (MPEG-PCL, MP) as biocompatible materials were also candidates for studies, as demonstrated by Park’s group [60], on Sprague Dawley rats subjected to acoustic trauma. In this study, they divided 43 Sprague Dawley rats into a control group and three case groups treated with IT DEX-phosphate (10 mg) 3 h after NE: group 1° was associated with a saline solution as the vehicle, group 2° with HA, and group 3° with MP.

They observed that all four groups improved their hearing thresholds after treatment; nevertheless, there was a difference in the degree. The HA + D group was significantly better than the MP + D group (*p* = 0.043). The difference between the HA + D group and the saline + D group was not statistically significant (*p* = 0.083). The duration of the vehicle/drug observed in the bulla was significantly longer for the MP + D group than for the saline + D and HA + D groups (*p* = 0.038).

Finally, many drugs have been used to improve the concentration in the RWM in order to increase its diffusion to the inner ear. In this way, caprate acid, a medium-chain fatty acid, is a well-recognized absorption enhancer that increases the permeability of macromolecules through the intercellular tight junctions of intestinal cells and also permeabilizes the blood–brain barrier.

In 2021, Jeong et al. [61] carried out a study in which they exposed Sprague Dawley rats to NE and treated them with IT DEX sodium phosphate (5 mg/mL) in combination with sodium caprate (SC) (1.94 mg/mL).

They demonstrated that the rat group treated with DEX + SC presented a significant increase in drug concentration in the cochlea perilymph 30 and 90 min after IT injection (*p* = 0.0255 and *p* = 0.0206, respectively).

Furthermore, they appreciated that SC cotreatment significantly accelerated the recovery from acoustic trauma 1 day after NE. Immunohistochemical analysis of the RWM exposed to SC also showed changes in intercellular spaces, and irregular contours were observed 30 min after sodium caprate administration.

#### 3.4.2. Anti-Inflammatory Not Glucocorticosteroids

The main results obtained in the review are presented in Table 3, and the most representative results are explained as below.

Blot analysis, indicating that the inhibition of NF-κB pathway might contribute to downstream inflammatory protein expression.
antioxidants-13-01105-t003_Table 3Table 3Characteristics of anti-inflammatory non-corticosteroids that were included in the systematic review and their main outcomes.ReferencesDesignPopulationAgeGenderTreatmentNoise ExposureType, Dose, and Duration of InterventionParameters AnalyzedMain OutcomesConclusions1. Chen (2023).Fujian (China) [62]Prospective cohortC57BL/6J mice5–6 weeksmaleOridonin (Ori)Broadband noise (120 dB SPL 0.2 kHz–70 kHz) for 4 hIP Ori (5 mg/kg) for 24 h, repeated daily for 14 daysa. ABR: +4 d, +7 d, and +14 db. RNA sequencingc. ImmunofluorescenceOri treatment (Ori + NE) had significant protective effects on the ABR threshold at all frequencies (*p* < 0.01 or *p* < 0.05). Transcriptional activation of the endogenous anti-inflammatory factor IL1R2 is an unreported anti-inflammatory mechanism of Ori and presents a novel strategy for treating sensorineural hearing loss2. Paciello (2020).Rome (Italy) [63]Prospective cohortWistar2 monthsmaleCaffeinic Acid (CA)Pure tone (120 dB SPL, 10 kHz) for 1 hIP injection of caffeic acid (30 mg/kg) daily for 3 daysa. ABR: 0 d, +1 d, +3 d, +7 d, and +21 db. Morphological analysis (*F*-acting staining +6 d)c. Oxidative stress (ROS/RNS and lipid peroxidation detection)d. Antioxidant defenses (Nrf2/HO-1 inmunofluorescence)e. Inflammatory process (*NF-kB* and *IL-1B*)a. Antioxidant effect: CA enhanced Nrf2 expression and its translocation into the nucleus, indicating the upregulation of HO-1 protein expression, reflecting the endogenous antioxidant response after NIHL b. Anti-inflammatory effect: CA effectively blocked the noise-induced increase in inflammatory molecules NF-kB and IL-1B3. Zawami (2016).Montreal (Canada) [64]Guinea pigs6 monthsfemalePure tone (110 dB SPL, 6 kHz) for 1 h on days +1 and +8IP injection caffeic acid (25 mg/kg) daily for 15 da. ABR: 0 d, +1 h, +4 h, +8 h, +11 h, and +15 hb. Morphological analysisa. In the NE group, ABRs showed the complete recovery of ATS by days 8 and 15 at all frequencies except 20 kHz. b. In the NE + CA group, ATS recovery was impaired at days 4 and 8 at all frequencies (*p* > 0.05). A daily dose of caffeine had a negative effect on hearing recovery after acoustic overstimulation at multiple frequencies4. Akil Ocal (2019).Ankara (Turkey) [65]Prospective cohortSprague Dawley11 monthsmaleoxytocinWhite noise (107 dB SPL) for 15 hIT injection of oxytocin (0.1–0.3 mL, it) on days: 1, 2, 4, 6, 8, and 10a. DPOAEb. ABR: 0 d, +1, +7, and +21 dc. Immunofluorescence (caspases 3, 8, and 9, and dUTP)In the NE + oxytocin group, ABR thresholds increased significantly on day 1 after acoustic trauma, but no significant differences were observed between thresholds at baseline and on days 7 and 21. Additionally, no significant differences were observed in distortion product otoacoustic emission signal-to-noise ratios measured before and on days 7 and 21 after acoustic traumaAbbreviations: NE (noise exposure). IT (intratympanic). ATS (auditory threshold shift). IP (intraperitoneal). ROS/RNS (reactive oxygen species).

##### Oridonin (Ori)

Inflammasomes, polyprotein complexes that sense intracellular and extracellular stimuli, are one of the mediators implicated in the inflammatory response of NIHL [63]. Regarding this inflammatory cascade, NOD-like receptor protein 3 (NLRP3) is one of the critical sensory molecules that mediates inflammasome maturation and the “cytokine storm” [66].

Oridonin provides otoprotection after NE by inhibiting the release of inflammasome-independent pro-inflammatory cytokines (TNFα and IL6) [67] by blocking the interaction with NLRP3. Additionally, Ori regulates the IL1R2 receptor, which may be another important otoprotective factor. IL1R2 is structurally similar to IL1R1; nevertheless, due to the lack of the Toll/IL1 receptor domain and because of its short cytoplasmic domain, it cannot interact with myeloid differentiation factor 88. Therefore, it cannot activate the inflammatory cascade, negatively regulating IL1 function [68]. In 2023, Chen et al. [62] proposed that Ori administration could enhance ILR2 concentration by specifically blocking the assembly and activation of NLRP3 inflammasomes, and thus having an anti-inflammatory effect on rats exposed to NIHL.

In order to demonstrate this, they treated C57BL/6J mice with Ori after NE. They proved that the immunofluorescence expression of NLRP3 in SGNs and IHCs, and OHCs decreased significantly (*p* < 0.01) in the cochlea of Ori-treated mice. They also noted that the average number of OHCs per visual field at the 14th day post-NE was significantly different (*p* < 0.05): 41 vs. 52 (NE vs. NE +Ori group). Furthermore, they caused a decrease in the concentration of p65 (pp65), NLRP3, IL1β, and IL1α, which was detected by Western

In order to study whether IL1R2 overexpression improves NIHL or not, Chen et al. [62] introduced a viral AAV vector overexpressing IL1R2 into the cochlear semicircular canals of the mice. They analyzed the overexpression of IL1R2 via immunofluorescence with the following results.

The difference of ribbon synapses per IHC in the Mock-AAV + NE-7d (control group) and IL1R2-AAV + NE-7d groups was highly significant (*p* < 0.01): 10.5 vs. 13, respectively. Although the average number of OHCs per visual field in the control group and IL1R2-AAV + NE-7d groups was positive, this difference was not significant: 48.9 vs. 50.7. They concluded that IL1R2 overexpression protects OHCs and ribbon synapses, especially ribbon synapses in IHCs.

##### Caffeic Acid (CA)

In 2020, Paciello et al. [63] exposed Wistar rats to pure tone noise for 1 h and treated them 1 h after NE with 30 mg/kg of IP CA for 3 days. They demonstrated that the NE + CA group had a statistically significant (*p* < 0.05) attenuation of the threshold shift for all frequencies in comparison to the NE group at all time points, but especially on day 7.

With respect to the morphological analysis, NE mainly affected OHCs. In the group treated with CA, hair cell survival reached higher levels in the middle basal turn compared to the NE group.

Paciello et al. also [63] assessed the cochlear oxidative stress caused by NE by measuring ROS/RNS and lipid peroxidation. They demonstrated that CA administration counteracted the imbalance of cochlear redox by reducing the amount of superoxide in all cochlear structures.

In terms of endogenous antioxidant defenses (Nrf2/HO-1), they also showed that CA treatment increased Nrf2 expression [63]. This enhancement of Nrf2 expression and its translocation into the nucleus following CA administration could be due to the upregulation of HO-1, indicating the activation of an endogenous antioxidant and cochlear protective pathway after noise injury. In contrast to these findings, Zawami et al. [64] conducted a study with guinea pigs in which they demonstrated that a daily dose of caffeine negatively affected hearing recovery after NE.

The same authors exposed guinea pigs to 110 dB SPL, 6 kHz for 1 h at two different times (days +1 and +8 of the experiment). They were treated via the IP administration of 25 mg/kg of CA daily for 15 days with the following results: While in group 2 (NE only) ABR showed the complete recovery of the threshold shift by day 8, in group 3 (NE + CA), threshold shift recovery was impaired at days 4 and 8 at all frequencies. In the last group, morphological analysis showed an abnormal arrangement of the Corti tunnel and the stria vascularis, with vessel dilation and microscopic bleeding.

##### Oxytocin

The anti-inflammatory and antioxidant effects of oxytocin have been well documented in the renal system. Rashed et al. [69] demonstrated the protective role of oxytocin against cisplatin nephrotoxicity in rats. On the other hand, Tuğ tepe et al. [70] observed the benefits of IP oxytocin in the recovery of renal architecture damage caused by renal ischemia reperfusion syndrome.

Nevertheless, to the best of our knowledge, no studies analyzing the otoprotective effect of oxytocin were published until Akil et al. [65] conducted theirs. In their study, they analyzed the otoprotective effect of IT oxytocin administered to Sprague Dawley rats exposed to white noise (107 dB SPL) for 15 h. 

In this study, they demonstrated that the oxytocin-treated group after NE showed a significant difference between ABR thresholds measured before NE and those measured on day 1 after NE. However, no statistically significant differences were observed between the ABR thresholds before NE and those on days 7 and 21 after NE (*p* = 0.564 and 0.655, respectively). In addition, they observed significant differences in signal-to-noise ratios (DPOAE) measured between the control group and the group treated with oxytocin, suggesting IT oxytocin treatment is beneficial for repairing NIHL, primarily when the damage involves OHCs.

### 3.5. Anti-Apoptotic Drugs

The most outstanding findings of the systematic review are shown in Table 4.
antioxidants-13-01105-t004_Table 4Table 4Characteristics of the anti-apoptotic drugs that were included in the systematic review and their main outcomes.ReferencesDesignPopulationAgeGenderTreatmentNoise ExposureType, Dose, and Duration of Intervention Parameters AnalyzedMain OutcomesConclusions1. Malfeld (2023). Hannover (Germany) [71]Prospective cohortGuinea pigs-maleInsulin-like Growth Factor 1 (IGF-1)Beethoven’s 5th Symphony, 4th movement (120 dB) for 4 h, 1 week after implantationContinuous administration of IGF-1 (0.3 ng/h) via implanted osmotic pump unilaterally a. ABR: −7 d, 0 d, +1 d, and +7 db. HistologyNo significant difference in threshold shifts between IGF-1 and AP implanted groups. No significant improvement in NIHL with preventive IGF-1 administration.2. Lin (2021). Taipei (Taiwan) [72]Prospective cohortGuinea pigs--IGF-1 MB exposed to ultrasound (US)Narrowband noise (118 dB SPL, 8 kHz) for 5 h200 µL MB exposed to US for 1 min (×3), followed by gelatin sponge soaked with 10 µL rhIGF-1 on the round window membrane (RWM) 24 h after noise exposurea. ABR: 0 d, +14 d, and +28 d b. Histologyc. RNA sequencing (*Akt1*, *MAPK1*, *3*)The USM group had the lowest threshold shift in ABR, the lowest loss of cochlear outer hair cells, and the lowest reduction in the number of synaptic ribbons on post-exposure day 28 among the three groups. The combination of USMBs and IGF-1 demonstrated a better treatment effect than IGF-1 alone.3. Cho (2021).Seoul (South Korea) [73]Prospective cohortSprague Dawley6 weeksMaleHA-TCO (1 mL) + IGF-1 (2.5 mg) + mD (5 mg)White noise (116 dB SPL) for 3.7 h.Intratympanic (IT) injection of 30–60 μL of the dual vehicle (HA-Tet + mDEX) + HTCA + IGF-1 + mDEXa. ABR: 0 d, +8 d, +12 d, 30 d, and 45 d b. Light ear endoscopyc. Fluorescence ear endoscopyd. Histologye. CT scana. The residence time of the drug/vehicle in the middle ear was extended 10.9 times using the dual vehicle, namely, cross-linked HA and PLGA microcapsules. b. The treatment outcome in terms of hearing threshold and hair cell count was comparably good in both groups.4. Xiaogang (2022). Shaanxi (China) [74]Prospective cohortC57 BL/6J mice6–8 weeksMaleLS19-Forskolin nanoparticles (LS19-FSK-NP)Broadband white noise (115 dB SPL, 2–20 kHz) for 2 hRound window membrane (RWM) injection of 10 μL LS19-FSK-NPa. ABR: 0 d, +1, +7, and +14b. Histologyc. ImmunofluorescenceThe sustained release and cumulative concentration of FSK in the cell inhibited the apoptosis of cochlear OHCs. LS19 peptide modification significantly improved the protective effect of LS19-FSK-NPs against NIHL based on ABR testing at 4, 8, 16, and 32 kHz.5. Shukla (2019).Delhi (China) [75]Prospective cohortSprague Dawley-MaleSelective adenosine (A2A) receptor agonist (CGS21680)White noise (100 dB SPL) 2 h daily over 15 daysIntraperitoneal (IP) injection of 100 μg/kg/day for 15 successive daysa. ABR: 0 d and +15 db. Cognitive assessment c. Histology d. ImmunofluorescenceATS post-exposure was 50.83 ± 4.26 dB in the noise group, 32.33 ± 4.5 dB in the drug group, and a threshold shift of 24.33 dB compared with the control group. The A2A receptor agonist CGS21680 provides protection from NIHL by maintaining hearing threshold levels and promoting neurogenesis in the hippocampus.6. Liu (2018). Mongolia (China) [76]Prospective cohortC57 BL/6J mice8 weeks-AK-796 dB SPL (8–16 kHz) for 12 h or 24 hIP injection of 30 mg/kg AK-7 1 day before noise exposure, then continuously with 15 mg/kg until the end of the experimenta. ABR: 0 d, +1 d, +7 d, and +14 db. Histology c. Immunostaining d. Western blotABR in the AK-7 group decreased significantly (*p* < 0.01) at days 1, 7, and 14 after NE. The SIRT2 inhibitor AK-7 reduces oxidative DNA damage and apoptosis in HEI-OC1 cells by suppressing the expression of caspase 3 and Bax, and by recovering the expression of Bcl-2.

#### 3.5.1. IGF-1

Insulin Growth Factor-1 (IGF-1) enhances anti-inflammatory and anti-apoptotic effects through the PI3K/AKT, ERK/MAPK and p38 or JNK signaling cascades [77]. In 2023, Malfeld et al. [71] administered IGF-1 via an osmotic pump into the cochlea of guinea pigs and exposed them to Beethoven’s 5th Symphony. This noise exposure of 120 dB for 4 h, 1 week after the implantation of IGF-1 in the cochlea and the round window membrane, resulted in inner ear protection against NIHL compared to the untreated animals with preserved OHCs in the basal and second turns. These results were corroborated by Lin and by Cho, both in 2021 [72,73], with other types of experiments where IGF-1 was administered; normal ABRs were obtained compared to the untreated group and hair cell loss was minimal [65].

#### 3.5.2. Forskolin (FSK)

Forskolin (FSK), a complex labdane diterpenoid, found in the root of a plant, Coleus forskohlii, was studied as an otoprotective agent by Xiaogang et al. in 2022 [61,74]. The authors designed a nanoparticle that included FSK and that specifically targeted a protein expressed in the membrane of OHCs in the inner ear. They concluded that the administration of this nanoparticle had a significant protective effect against NIHL 1, 7, and 14 days, improving auditory functions in different thresholds.

#### 3.5.3. Selective Adenosine (A2A) Receptor Agonist (CGS21680)

Adenosine, an endogenous neuromodulator, is another anti-apoptotic molecule that enhances the endogenous antioxidant system and downregulates glutamate, triggering anti-inflammatory response and promoting angiogenesis [78]. Moreover, the increase in endogenous adenosine has been associated with the protection of the organ of Corti against noise. In this context, Shukla et al. [75] performed a study in which A2A was injected intraperitoneally (IP) into mice exposed to 100 dB of white noise for 15 days, 2 h per day. From this experiment, they concluded that, in the noise-exposed group, the OHCs were intact in the treated group compared to the control groups, in which these cells were damaged, results that correlated with the better ABR threshold pattern in the A2A group.

#### 3.5.4. AK-7

Finally, Liu et al. [76] compared the effect of 3-(1-azepanylsulfonyl)-N-(3-bromphenyl) Benzamide (also referred to as AK-7), a specific Sirtuin 2 (Sirt2) inhibitor, in mice to demonstrate the effect of the sirtuin inhibitor on the mice’s hearing after sound exposure. Sirt2 is a NAD-dependent protein deacetylase that regulates oxidative stress [79]. In this study, they demonstrated that AK-7-treated mice showed a reduced expression of proapoptotic proteins 1, 7, and 14 days after noise exposure, such as caspase-3 and BAX, and an increased expression of the anti-apoptotic protein BCL-2. From these results, they concluded that AK-7 administration could protect mice cochlea from noise-induced hair cell death. 

### 3.6. Application of the Included Pharmacological Therapies in Human Beings

Of the 41 articles selected for this systematic review, 37 corresponded to studies whose drugs were used in experimental animals, while only 4 experimented with drugs on humans. These 4 studies were carried out on a cohort of Iranian textile workers on whom NAC was used [29], on personnel of the Swedish armed forces also using NAC [33], on a cohort exposed to gunshot noise on which PD was used [35], and on a group of South Korean soldiers in which PD+ DEX was used [34].

However, although the rest of the 37 included studies did not experiment with drugs in humans, the FDA and the EMA approved some of the drugs included in this review for human use, demonstrating their effectiveness and safety. Below, the pharmacological therapies included in this review are ordered according to whether or not they are approved for human use by the FDA and EMA.

#### 3.6.1. Approved as a Drug for Use in Humans by the FDA and EMA

1.NAC [80]: first approved for use on humans in 1899 as a mucolytic agent, thanks to the mucolytic effect of its free sulfhydryl group, which acts by breaking the disulfide bridges of mucoproteins, decreasing the viscosity of mucus.2.D-methionine [81]: approved for use in humans as an IV serum to reconstitute amino acids. The most marketed form of the drug, a 500 mL unit provides a total of 40 g of amino acids, of which 0.10 g corresponds to Methionine.3.Fluvastatin [82]: approved in 1993 for use in the treatment of heterozygous familial hypercholesterolemia (heFH) in pediatric patients.4.Resveratrol [83]: pending approval as a drug for the treatment of ALS. Pending closure of an RCT comparing the combined treatment of resveratrol with dutasteride versus riluzole for the treatment of patients with ALS.5.Hydrocortisone [84]: approved in 1952.6.Prednisone [85]: approved in 1955.7.Methylprednisolone [86]: approved in 1957.8.Dexamethasone [87]: approved in 1961.

Human experience with corticosteroids (5–8) is very extensive and very safe. So much so that corticosteroids today have approval from the FDA/EMA for numerous diseases. Specifically, they have approval for 11 different types of disease categories [e–h].
9.Oxytocin [88]: approved in 1959 for labor induction.10.IFG [89]: Approved in 2005 for use in the treatment of severe growth deficiencies in children with certain genetic conditions, such as growth hormone deficiency syndrome.

#### 3.6.2. Approved as a Multivitamin Supplement or Herbal Medicinal Product by the FDA and EMA

Rosmarinic acid [90]: approved in 2021 as an antioxidant component for the treatment of dementia and Alzheimer’s.Berberine [91]: Approved as an antioxidant homeopathic medicine.Ginkgo biloba [92]: approved in 2015 by the EMA as a therapy to improve the age-related cognitive impairment and quality of life of adults with mild dementia.Coenzyme Q10 [93]: approved as a homeopathic antioxidant medicine without therapeutic indications.Caffeic acid [94]: used for its anti-inflammatory action for the temporary relief of sinus congestion, headache, indigestion, joint pain, and dizziness. It has 3 RCTs in development.LS19-Forskolin [95]: Forskolin is a compound extracted from the Coleus forskohlii plant and has been studied for its possible effects on weight loss, glaucoma control, and other potential uses.

#### 3.6.3. Not Approved by FDA or EMA and without RCTs in PubMed

Hk-2 [96]: there are two cohort studies in PubMed as a therapy to prevent NIHL.SOD ZIF-8 [97]: one cohort study is recorded in PubMed.IDB [98]: there are two cohort studies in PubMed.Oridonin [99]: two cohort studies.CGS 21680 [100]: two cohort studies.AK-7 [101]: one cohort studies.

#### 3.6.4. Possible Use of Therapies as an Otoprotective Drug

Despite most of the proposed drugs that have been approved by the FDA/EMA for their use in diverse pathologies, some of them, such as oxytocin, have been associated with hormonal effects and other secondary effects, such as diarrhea or nausea. Dietary supplements and medications, like NAC, resveratrol, or multivitamins have been demonstrated to provide beneficial effects preventing cell damage, reducing inflammation, or reducing symptoms in psychiatric disorders. However, most of them may cause some risks, mostly if the dose is not regulated. Therefore, to use them as otoprotective drugs, it is important to personalize therapies for each patient, in order to define the correct drug, timeline, and dose, and reduce the secondary effects.

## 4. Discussion

Several studies have demonstrated the effectiveness of antioxidants, anti-inflammatories, or anti-apoptotics drugs for the prevention and treatment of NIHL. However, the administration mode, the dose, and the noise exposure methods have not been completely elucidated yet for humans. Although nowadays there is no method approved by the FDA and EMA as exclusive pharmacological therapy for the treatment of NIHL, many of the drugs analyzed in this systematic review were approved long ago for the treatment of multiple diseases in humans. In fact, many of the dosages and routes of administration proposed in this study have been used for years in humans, with excellent results in terms of safety and efficacy.

An example of the use of these drugs in humans for a long time and proposed as therapy in this study would be: oral NAC 100 mg, which is widely used as an antiplatelet agent [80], as proposed by Ada [39]; oral PD at 60 mg, universally used as an anti-inflammatory drug [85], as suggested by Chang and Choi [34,35]; the IT injection of DEX at 0.1 mL, approved for the treatment of sudden neurosensory deafness [87], as recommended by Heinrich and Shih [55,59]’ or the IV injection of Hydrocortisone 0.05%, as proposed by Panevin [53]. All these therapeutic possibilities make us think that the pharmacological therapies that we include in this review might be successfully applied to treat noise induction hearing loss in the near future.

So, the mechanism through which the variety of these treatments work in the hearing pathway as well as their ability to penetrate the blood–brain barrier (BBB) are relevant issues to study in order to consider them as valid therapy for patients. Several substances have been considered as therapeutic targets for central nervous system diseases, but their limited ability to penetrate the BBB, as in the case of glutathione, which only penetrates 1% when administered exogenously [54], reduces the effectiveness of mitigating oxidative stress in the central nervous system.

Moreover, the enrollment of patients affected by environmental noise exposure that can be possibly treated with antioxidants, anti-inflammatories, or anti-apoptotic drugs is particularly limited due to the short length of the therapeutic window and the long period needed to observe treatment results [23]. Additionally, factors such as subclinical symptoms and individual variations in noise susceptibility increase the difficulties to recruit participants. This is the reason why most of the studies regarding possible protective factors against NIHL have been performed on animal models, such as mice, chinchilla, or guinea pig.

It has been concluded from experimental studies that early onset therapy confers protection against NIHL, being an important point in the timeline for the drug’s administration. Strong data support the protective effects of antioxidants administered before and immediately after exposure to extreme noise. Among these studies, Campbell et al. [37] revealed a significant reduction in hearing loss when therapy was administered just 4 h after acoustic trauma, but not after 12 h, as well as D-met administration that provides hair cell rescue when administered just after noise exposure [38]. Other authors claim that the prophylactic administration of antioxidants also provides otoprotection from NE when administered days to hours before exposure [38,50]. However, prophylactic administration is difficult in patients because it is not considered as a standard measure in current protocols. But, prophylactic therapy can be considered, especially in the case of patients exposed to occupational noise, such as in military or construction settings, in which treatment can be anticipated. Less data exist regarding the efficacy of delayed treatment for NIHL in humans. Yamashita et al. [16] found that ROS and RNS persisted in hair cells for up to 7–10 days after acoustic trauma. Consequently, they described that the greatest window of opportunity was within 3 days of exposure, with a combination of antioxidants with different mechanisms of action and prolonging the duration of treatment.

Increased ROS production can lead to an inflammatory response in the hair cells, so the administration of anti-inflammatory molecules can be regarded as a preventive therapy for NIHL. In this sense, Prednisone and Dexamethasone, the most studied glucocorticosteroids, have been used in humans in order to treat acoustic trauma, and both have demonstrated hearing protection against inflammation [54,55]. Other non-glucocorticosteroid drugs have been studied to treat the inflammasomes, the inflammatory cascades that occur after NIHL and provoke cytokine release [58,59,60,61]. However, the timeline of their action is still unknown.

Finally, anti-apoptotic drugs, such as IGF-1 or AK-7, have been also linked to better hearing structure maintenance after noise exposure. From these results, the authors conclude that anti-apoptotic treatment administration can be protective 1, 7, and 14 days after noise exposure, with a reduction in pro-apoptotic protein expression, and that these substances enhance the endogenous antioxidant system in order to reduce the oxidative stress and inflammation that occur after acoustic trauma [65,66,67,68,69,70,71,72,73,74,75,76,77,78,79].

Ultimately, there is a need for more conclusive evidence demonstrating the protective effects of antioxidant/anti-inflammatory or anti-apoptotic drug administrations, the timeline in which they work, and the dose in which they should be used, to consider them as therapeutic agents. Moreover, the early diagnosis and subsequent treatment to increase the efficacy of these therapies must be taken into account.

## 5. Conclusions

Further studies are needed to fully understand the potential of the abovementioned antioxidants/anti-inflammatory and anti-apoptotic drugs as they may be a promising option to prevent and treat NIHL. Although there is convincing evidence that these compounds have beneficial anti-inflammatory or anti-apoptotic properties, there are still challenges to overcome the implementation of these treatments. The success of these therapies depends on factors such as the type and bioavailability of the drug, dosage, route of administration, and the timing and duration of therapy. Therefore, the next challenge is to adjust the optimal dosage and route of administration of these antioxidants to obtain the best results in patients. With this aim, additional, large, randomized, double-blind, placebo-controlled human trials are needed.

## Figures and Tables

**Figure 1 antioxidants-13-01105-f001:**
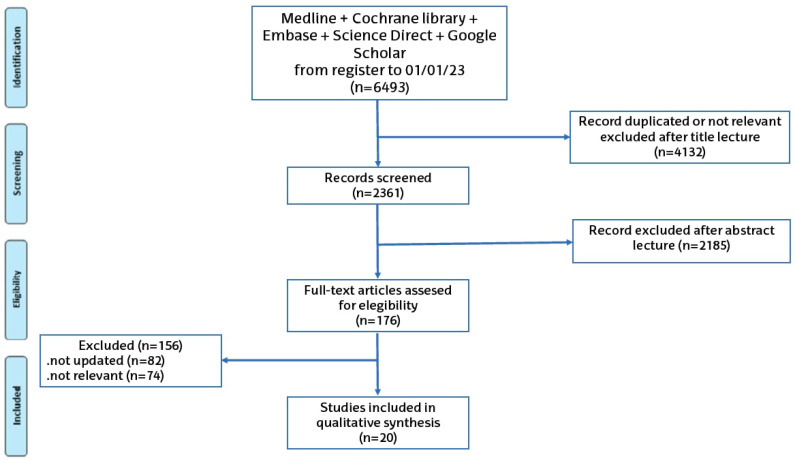
Systematic review strategy flowchart.

**Figure 2 antioxidants-13-01105-f002:**
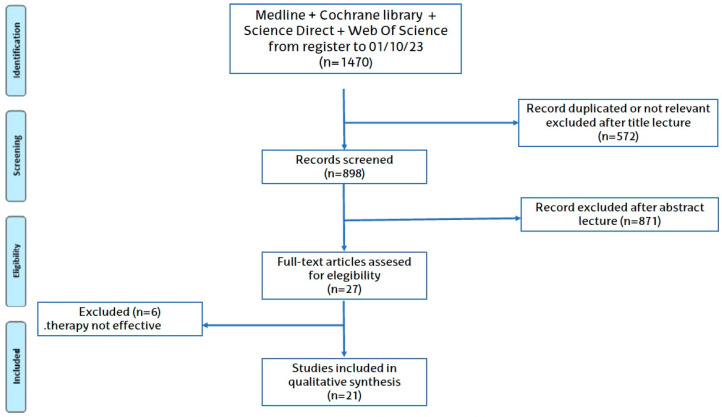
Systematic review strategy flowchart.

## Data Availability

The data can be made available upon request. Email: miren.revuelta@ehu.eus.

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
