# Peer review of "Pathogenesis and New Pharmacological Approaches to Noise-Induced Hearing Loss: A Systematic Review"

_antioxidants, 2024, doi:10.3390/antiox13091105_

Round 1

Reviewer 1 Report

The review by Santaolalla uses PRISMA criteria to broach the subject of tharepies to prevent noise-induced hearing loss. I have no issues with the introduction and methodology. However, I believe that the therapies are not described with the proper focus. Indeed, authirs provide a comprehensive list of known possible therepies for NIHL, yet many of them are unpractical for human use (but they do not indicate which). Prevention might be the best way to use these therapies in people professionally exposed to high levels of noise, but this is nowhere discussed. Chronic versus acute noise exposure is not usually detailed in the different therapies described, and this is another important point to bear in mind. In all, I feel that a large amount of work has been devoted to reading all the relevant literature, but all the knowledge garnered has not been completely digested.

The text will greatly benefit from revision by a native English speaker to remove typos (e.g. "prednisona") and grammar errors.

Author Response

The review by Santaolalla uses PRISMA criteria to broach the subject of tharepies to prevent noise-induced hearing loss. I have no issues with the introduction and methodology. However, I believe that the therapies are not described with the proper focus. Indeed, authirs provide a comprehensive list of known possible therepies for NIHL, yet many of them are unpractical for human use (but they do not indicate which). Prevention might be the best way to use these therapies in people professionally exposed to high levels of noise, but this is nowhere discussed. Chronic versus acute noise exposure is not usually detailed in the different therapies described, and this is another important point to bear in mind. In all, I feel that a large amount of work has been devoted to reading all the relevant literature, but all the knowledge garnered has not been completely digested.

Major comments

  1. Indicate de applicability of the pharmacological therapies include in the systematic review for human use.

                        -material and methods: lines 122-130

                        -results: lines 437-485

                        -discussion: lines 488-500

  1. Comment the role of the prevention of the noise exposure.

                        -introducción: lines 83-87

                        -discussión: lines 498-500

  1. Detail in the different therapies described the noise exposition (acute vs chronic).

                        -results: see “Table 1”, “Table 2”, “Table 3”, “Table 4”

Detail comments:

The text will greatly benefit from revision by a native English speaker to remove typos (e.g. "prednisona") and grammar errors

                        -Text revised

Reviewer 2 Report

Interesting topic. Few, although consistent, revisions are required.

The topic of the article is interesting and the English language is correctly used. Please elucidate the following points:

  1. Specify and detailed why different databases has been used to perform the search.
  2. Please remove the red lines below the sentences in each boxes of figure 1 and 2.

3.       Please provide a table with the reference, the proposed therapy and the outcomes for each article.

Author Response

Major comments

  1. Specify and detailed why different databases has been used to perform the search

                        -material and methods: (lines 90-91)

  1. Please remove the red lines below the sentences in each boxes of figure 1 and 2.

                        -material and methods: done (figure 1 and 2)

  1. Provide a table with the reference, the proposed therapy and the outcomes for each article.

                        -results: see “Table 1”, “Table 2”, “Table 3”, “Table 4”

Reviewer 3 Report

In the manuscript "Pathogenesis and new pharmacological approaches to noise-induced hearing loss," the authors analyzed the effects of different pharmacological treatments, focusing on exogenous antioxidants, anti-inflammatories, and anti-apoptotics to reduce cellular damage caused by acoustic trauma in the inner ear. While the reviewer appreciates the authors' efforts, there are many grammatical mistakes and typographical errors throughout the manuscript, which make the quality of the manuscript unsatisfactory. The reviewer pointed out some, but not all, of them as "Minor points." The reviewer suggests that the authors have the manuscript externally edited by someone familiar with the field. Additionally, there are some major points that the reviewer would like the authors to address.

Major points:

1.In the manuscript, the authors mentioned that some results are summarized in tables, for example, "Table 1-4" in Line 202, "Table 1" in Line 213, "Table 2-3" in Line 332, "Table 4" in Line 474. However, the tables are not inserted in the manuscript. Please insert them.

2.In the "5. Conclusions" section, the authors mentioned that "Further studies are needed" and "the next challenge is to adjust the optimal dosage and route of administration of these antioxidants so that greater protective effects can be exerted in the human brain. With this aim, additional large, randomized, double-blind, placebo-controlled human trials are needed." Do the authors mean that adjusting the optimal dosage and route of administration of these antioxidants should be examined in humans?

Minor points:

- Line 21: Please insert "by" between "cellular damage" and "acoustic trauma."

- Line 24: The phrase "apoptotic drugs drug administration" is odd. Please revise it.

- Line 72: Please insert "of" between "Effect" and "apoptotic."

- Line 260: There are two periods after "to distribute it."

- Line 286: There are two periods after the sentence. Please remove one of them.

- Line 296: Do the authors mean "contralateral ear after noise exposure" instead of "contralateral ear ater noise exposure"?

- Line 297: Do the authors mean "determine" instead of "determin"?

- Line 310: Do the authors mean "using" instead of "usign"?

- Line 375: Please remove the "s" after the period.

- Line 397: There are two periods after the sentence. Please remove one of them.

- Line 400: Please insert a space between "acoustic" and "1."

- Line 424: Do the authors mean "detected" instead of "detecteed"?

- Line 445: There is a duplication of "the" before "up-regulation." Please remove one of them. Additionally, the phrase "indicating an endogenous antioxidant, cochlear protective pathway after noise injury" is odd. It is suggested to insert a verb.

- Line 462: The authors mentioned "white noise white." Is this correct English? If not, please revise it.

- Line 479: The authors mentioned "to to Beethoven." "To" is duplicated; please remove one of them.

- Line 548: The expression "effects of antioxidant/anti-inflammatory or anti-apoptotic drugs drug administration" is odd. Please revise it to a grammatically correct expression.

Author Response

Major comments:

  1. There are many grammatical mistakes and typographical errors throughout the manuscript, which make the quality of the manuscript unsatisfactory

            -text revised

Detail comments:

  1. Insert the tables

                        -results: see “Table 1”, “Table 2”, “Table 3”, “Table 4”

  1. “Conclusions" section, the authors mentioned that "Further studies are needed" and "the next challenge is to adjust the optimal dosage and route of administration of these antioxidants so that greater protective effects can be exerted in the human brain. With this aim, additional large, randomized, double-blind, placebo-controlled human trials are needed." Do the authors mean that adjusting the optimal dosage and route of administration of these antioxidants should be examined in humans?

-this has been specified and included all treatments that has been previously tested in humans.  Discussion 488-500.

Minor points:

- Line 21: Please insert "by" between "cellular damage" and "acoustic trauma."

            (OK: line 18)

- Line 24: The phrase "apoptotic drugs drug administration" is odd. Please revise it.

            (OK: line 21)

- Line 72: Please insert "of" between "Effect" and "apoptotic."

            (OK: line 68)

- Line 260: There are two periods after "to distribute it."

            (OK: line 209)

- Line 286: There are two periods after the sentence. Please remove one of them.

            (OK: line 253)

- Line 296: Do the authors mean "contralateral ear after noise exposure" instead of "contralateral ear ater noise exposure"?

            (OK: line 241)

- Line 297: Do the authors mean "determine" instead of "determin"?

            (OK: line 242)

- Line 310: Do the authors mean "using" instead of "usign"?

            (OK: line 252)

- Line 375: Please remove the "s" after the period.

            (OK: line 311)

- Line 397: There are two periods after the sentence. Please remove one of them.

            (OK: line 331)

- Line 400: Please insert a space between "acoustic" and "1."

            (OK: line 334)

- Line 424: Do the authors mean "detected" instead of "detecteed"?

            (OK: line 360)

- Line 445: There is a duplication of "the" before "up-regulation." Please remove one of            them. Additionally, the phrase "indicating an endogenous antioxidant, cochlear     protective pathway after noise injury" is odd. It is suggested to insert a verb.

            (OK: line 381 and 382)

- Line 462: The authors mentioned "white noise white." Is this correct English? If not,           please revise it.

            (OK: line 397)

- Line 479: The authors mentioned "to to Beethoven." "To" is duplicated; please remove            one of them.

            (OK: line 412)

- Line 548: The expression "effects of antioxidant/anti-inflammatory or anti-apoptotic drugs drug administration" is odd. Please revise it to a grammatically correct            expression.

            (OK: line 537)

Round 2

Reviewer 1 Report

I thank the authors for revising the manuscript following my advice. However, I still miss some discussion in section 3.5 about the possible use of approved therapies. Indeed, oxytocin is approved for human use and has shown some otoprotective effects, yet it is not a feasible therapy because of its hormonal effects. Occupational (i.e. airport workers, industrial jobs, etc) and recreational (i.e. concerts) exposure to noise is a field where investigation of prophylactic use of otoprotective drugs is bound to make an impact. This should be discussed, in my opinion.

Although most typos and minor English mistakes have been corrected, some new ones have appeared. Please revise the text again.

There are no additional comments

Author Response

Major comments

I thank the authors for revising the manuscript following my advice. However, I still miss some discussion in section 3.5 about the possible use of approved therapies. Indeed, oxytocin is approved for human use and has shown some otoprotective effects, yet it is not a feasible therapy because of its hormonal effects. Occupational (i.e. airport workers, industrial jobs, etc) and recreational (i.e. concerts) exposure to noise is a field where investigation of prophylactic use of otoprotective drugs is bound to make an impact. This should be discussed, in my opinion.

This has been discussed in the section 3.6.4.

Although most typos and minor English mistakes have been corrected, some new ones have appeared. Please revise the text again.

The whole manuscript has been revised by an editing service

Reviewer 2 Report

New version approved

New version approved

Author Response

thank you for your effort and acceptance.

Reviewer 3 Report

The reviewer appreciates that the authors have addressed the comments and revised the manuscript. However, grammatical mistakes and typographical errors are still present. Again, the reviewer strongly suggests that the manuscript undergo language editing by a native English speaker or professional editor. In the acknowledgments section, please indicate the name of the English proofreading company the authors used for this manuscript.

Some unsuitable expressions are as follows:

[1] In Tables 1, 2, 3, and 4, the title for the second column from the left is “Desing.” The reviewer does not understand the word “Desing.” Do the authors mean “Design”?

[2] Line 53: The sentence “Mitochondria is involved in metabolic damaged caused by acoustic trauma” is odd. Do the authors mean “damages” instead of “damaged”?

[3] Line 56: Do the authors mean “major” instead of “mayor”?

[4] Line 63: Do the authors mean “cytochrome c” instead of “c cytochrome”?

[5] Line 90: The sentence “And in order ~ were sought” is odd. What is the subject?

[6] Line 125: Do the authors mean “human beings” instead of “humans beings”?

[7] Line 137: The word “approximation” is a noun.

[8] Line 137: Do the authors mean “whole” instead of “hole”?

[9] Line 140: Since “inflammatory” is an adjective, a noun should be inserted after “inflammatory.”

[10] Line 141 and throughout the manuscript: Since “apoptotic” is an adjective, a noun should be inserted after “apoptotic.” Additionally, this subtitle is shown in uppercase letters in Line 405, while others are not. Please unify the method of listing.

[11] In the abbreviations of Table 1, a semicolon should be inserted between RCT and PTS.

[12] Line 224: Please check the grammar.

[13] Lines 285 and 286: Please insert a space between the number and “nm.”

[14] Line 402: Do the authors mean “DPOAE” instead of “DOPE”?

[15] The word “KHz” should be revised to “kHz” throughout the manuscript.

Author Response

thanks for your comments, the following unsuitable expressions have been corrected and the whole manuscript has undergo language editing by AJE (American Journal Experts):

[1] In Tables 1, 2, 3, and 4, the title for the second column from the left is “Desing.” The reviewer does not understand the word “Desing.” Do the authors mean “Design”?

            .done (Tables 1, 2, 3 and 4)

[2] Line 53: The sentence “Mitochondria is involved in metabolic damaged caused by acoustic trauma” is odd. Do the authors mean “damages” instead of “damaged”?

            .done (line 53)

[3] Line 56: Do the authors mean “major” instead of “mayor”?

            .done (line 55)

[4] Line 63: Do the authors mean “cytochrome c” instead of “c cytochrome”?

            .done (line 62)

[5] Line 90: The sentence “And in order ~ were sought” is odd. What is the subject?

            .done (line 89-90)

[6] Line 125: Do the authors mean “human beings” instead of “humans beings”?

            .done (line 124)

[7] Line 137: The word “approximation” is a noun.

            .done (line 136)

[8] Line 137: Do the authors mean “whole” instead of “hole”?

            .done (line 136)

[9] Line 140: Since “inflammatory” is an adjective, a noun should be inserted after “inflammatory.”

            .done (line 139)

[10] Line 141 and throughout the manuscript: Since “apoptotic” is an adjective, a noun should be inserted after “apoptotic.” Additionally, this subtitle is shown in uppercase letters in Line 405, while others are not. Please unify the method of listing.

            .done and unified (lines 140, 495, 102, 108, 135 and 486)

[11] In the abbreviations of Table 1, a semicolon should be inserted between RCT and PTS.

            .done (Table 1)

[12] Line 224: Please check the grammar.

            .done (line 223)

[13] Lines 285 and 286: Please insert a space between the number and “nm.”

            .done (line 283, 284)

[14] Line 402: Do the authors mean “DPOAE” instead of “DOPE”?

            .done (line 400)

[15] The word “KHz” should be revised to “kHz” throughout the manuscript

            .done throughout the manuscript